# Fungal Enzymes Involved in Plastics Biodegradation

**DOI:** 10.3390/microorganisms10061180

**Published:** 2022-06-08

**Authors:** Marta Elisabetta Eleonora Temporiti, Lidia Nicola, Erik Nielsen, Solveig Tosi

**Affiliations:** 1Laboratory of Mycology, Department of Earth and Environmental Sciences, Università degli Studi di Pavia, Via S. Epifanio 14, 27100 Pavia, Italy; lidia.nicola@unipv.it (L.N.); solveig.tosi@unipv.it (S.T.); 2Department of Biology and Biotechnology, Università degli Studi di Pavia, Via Ferrata 9, 27100 Pavia, Italy; erik.nielsen@unipv.it

**Keywords:** plastic, biodegradation, enzymes, fungi, bioremediation, biotechnology

## Abstract

Plastic pollution is a growing environmental problem, in part due to the extremely stable and durable nature of this polymer. As recycling does not provide a complete solution, research has been focusing on alternative ways of degrading plastic. Fungi provide a wide array of enzymes specialized in the degradation of recalcitrant substances and are very promising candidates in the field of plastic degradation. This review examines the present literature for different fungal enzymes involved in plastic degradation, describing their characteristics, efficacy and biotechnological applications. Fungal laccases and peroxidases, generally used by fungi to degrade lignin, show good results in degrading polyethylene (PE) and polyvinyl chloride (PVC), while esterases such as cutinases and lipases were successfully used to degrade polyethylene terephthalate (PET) and polyurethane (PUR). Good results were also obtained on PUR by fungal proteases and ureases. All these enzymes were isolated from many different fungi, from both *Basidiomycetes* and *Ascomycetes*, and have shown remarkable efficiency in plastic biodegradation under laboratory conditions. Therefore, future research should focus on the interactions between the genes, proteins, metabolites and environmental conditions involved in the processes. Further steps such as the improvement in catalytic efficiency and genetic engineering could lead these enzymes to become biotechnological applications in the field of plastic degradation.

## 1. Introduction

The word plastic derives from the Greek “plastikos”, meaning “able to be modeled” [1]. Today, the term plastic refers to a range of synthetic long-chain polymeric molecules which, in the 1950s, started to be substituted for natural materials across a range of different sectors and in everyday applications [2]. The rapid development of plastics can be attributed to their combination of lightness, durability and other intrinsic properties, along with their easy and low-cost production [3]. As a result of their versatility, plastic materials have been increasingly used, reaching global production of almost 370 million tonnes in 2020 [4]. Almost 55 million tonnes of plastic were produced in Europe in 2020 [4]. The most commonly used plastics are polyethylene (PE; 30.3%), polypropylene (PP; 19.7%), polyvinyl chloride (PVC; 9.6%), polyethylene terephthalate (PET; 8.4%), polyurethane (PUR; 7.8%) and polystyrene (PS; 6.1%) [4,5].

### 1.1. Plastic Pollution

The mass of plastics in municipal solid waste in high-income and developing countries increased from less than 1% in 1960 to more than 10% in 2005 [6]. In Europe, 6.9 million tonnes of plastics were dumped in landfills in 2020 [4]. One of the problems of plastics accumulated in landfills or released into the environment, is the long time they take to decay. This long decay time derives from the inherent characteristics of plastic, especially the high molecular weight, crystallinity and hydrophobicity [3,7], and from the fact that its monomers, such as ethylene and propylene, originate from fossil hydrocarbons [8]. This results in the accumulation and persistence of plastic in land, freshwater, and oceans for many decades [9,10]. Moreover, supplementary chemicals and additives are often added to plastic polymers to increase the quality of the final products [11]. These additives such as endocrine disrupting chemicals (bisphenol A, bisphenol S, octylphenol and nonylphenol) [12], dioxin-like compounds [13] and heavy metals [14,15] can cause negative effects on organisms. Reproductive abnormalities, disruption of the endocrine system, diabetes and obesity could be linked to additives in plastics [16].

A further problem associated with plastic pollution is the formation of small particles called microplastics, which originate from plastic fragmentation. Microplastics and bigger plastic fragments can enter the food chain and be transferred to higher trophic level organisms where they accumulate [17]. Microplastics can also be the carrier for toxic chemicals and pathogens, facilitating their dispersion in the environment and threatening ecosystems [3,5].

In 2020, 12.4 million tonnes of plastics were used for energy recovery [4] and plastic incineration plays an important role in the management of municipal solid waste [18]. However, energy recovery by incineration can lead to harmful and toxic emissions, such as dioxins, furans, heavy metals and sulphides, contributing to environmental pollution [19,20,21]. Recycling is a better alternative for plastic waste management, but it is not the ultimate solution to the plastic problem. For example, the mechanical properties of recycled PET are reduced with each reuse and the thermal degradation of PE by pyrolysis leads to the random breaking of C–C bonds with the consequent drastic change in its mechanical properties [22]. Moreover, the extremely low percentage of PP recycling (<1%) is alarming since in many cases it is not found as the only polymer forming an object. In addition, the tertiary carbon in PP is susceptible to photo-oxidative and thermo-oxidative degradation, requiring a stabiliser to be added in the production stage, contributing to the deterioration of recycled PP properties [22].

### 1.2. Plastic Biodegradation

Biodegradation is a complex process of physico-chemical transformation of polymers into smaller units mediated by microorganisms [23,24]. Microorganisms, including fungi, are able to biochemically degrade, assimilate and metabolise complex organic compounds, xenobiotics and recalcitrant substances for their energy needs [25,26]. Several organisms and different mechanisms are being investigated at the moment to improve and promote the biodegradation of complex and polluting polymers. For example, the addition of bacteria with specific engineered plasmids in polluted sites could transfer the catabolic genes in the plasmids to the indigenous bacterial population increasing their capacity for xenobiotic degradation [27]. Furthermore, it is possible to insert mutations in bacterial genes that increase the biodegradative capacity of their enzymes. Thanks to this technique, Lu et al. [28] created an algorithm for an engineered and robust PET hydrolase that can be active in a wide range of temperatures and pHs. Another interesting line of research is to exploit extremophile microorganisms, so that they can be used in the bioremediation of extremely polluted sites. For example, some fungi, such as strains belonging to *Fusarium*, *Verticillium*, *Penicillium* and *Aspergillus*, are able to produce metal nanoparticles that allow them to tolerate and remove heavy metals from heavily polluted water or soil [29].

The first step required for the biodegradation of high molecular weight and long-chain polymers, such as plastics, is the weakening of the polymers’ structure. Many different factors can influence plastic biodegradation, for example, the hydrophobicity of the exposed area as well as the chemical structure, crystallinity grade and structure, glass transition, melting temperature and elasticity [30,31,32]. Environmental factors, such as UV exposure or temperature, can decrease the hydrophobicity of plastics or introduce carbonyl/carboxyl/hydroxyl groups, increasing their biodegradability [33,34,35]. These environmental factors can lead to surface roughness, cracks and molecular changes in plastics [36]. Similarly, the growth of microorganisms on plastic surfaces can modify the physical properties of the plastic by creating cracks and enlarging the pore size. Microorganisms can also chemically deteriorate plastics, for example, by changing the pH of the surrounding microenvironment [37].

The second step in plastic biodegradation is the depolymerisation into shorter chains. The microbial exoenzymes involved in this process create intermediates with modified properties, that increase their cellular assimilation [38]. After intermediates are created and assimilated, they are used by cells as carbon sources and broken down into water and carbon dioxide or methane to complete the mineralization process [5,24].

The complexity of plastic biodegradation is due to their chemical and physical characteristics, such as their high molecular weight, hydrophobicity and insolubility [39]. The use of filamentous fungi for the bioremediation process of plastics can overcome this problem. Indeed, filamentous fungi present a typical hyphal apical growth form that allows them to extend their mycelial networks into different kinds of materials [40]. The penetrative abilities of fungal hyphae are associated with their secretion of exoenzymes and hydrophobins, increasing their adhesion to hydrophobic substrates [41]. Moreover, the non-specificity of fungal exoenzymes allows them to break down different plastic polymers [42]. For example, fungal hydrolases (lipases, carboxylesterases, cutinases and proteases) can modify the plastic surface, increasing its hydrophilicity [43]. These enzymes are also involved in PET and PUR biodegradation due to the presence of hydrolysable chemical bonds in the polymer structures [44,45,46]. On the other hand, oxidoreductases (laccases and peroxidases) are involved in plastic degradation into smaller molecules such as oligomers, dimers and monomers [47,48]. Due to their highly stable carbon–carbon (C–C) bonds, plastic polymers such as PE, PS, PP and PVC require oxidation before the depolymerisation process [39,49].

In this context, this review will examine the different fungal enzymes involved in the degradation processes of the primary petroleum-based plastic polymers, describing their main characteristics, their efficacy and their possible biotechnological applications. Therefore, the aim of this review is to provide an extensive and reliable assessment of all the present knowledge on fungal enzymes in plastic biodegradation as a start base for new bioremediation applications.

## 2. Fungal Enzymes Involved in Plastic Biodegradation

The main classes of enzymes involved in plastic biodegradation are hydrolases and oxidoreductases (Table 1). These enzymes have been extensively studied due to their involvement both in natural and industrial processes. For example, in nature and industry they are essential in lignocellulose biodegradation [50], fungal pathogenesis [51] and the hydrolysis of triacylglycerol to fatty acids [52]. These enzymes also have applications in the food and textile industries [53,54,55], and in bioremediation processes [56,57].

### 2.1. Laccases (EC 1.10.3.2)

Laccases (EC 1.10.3.2) are a class of enzyme belonging to the blue copper oxidases and are multicopper monomeric glycoproteins [58]. They use oxygen as an electron acceptor to oxidize phenolic and non-phenolic compounds, and they are involved in the reduction of molecular oxygen to water [59,60,61].

Laccases were first discovered in the plant species *Rhus vernicifera* in 1883 [62]. In 1896 they were identified in fungi for the first time by Bertrand and Laborde [63,64]. In the following years, laccases were discovered in many species of fungi belonging to *Ascomycetes*, *Basidiomycetes* and *Deuteromycetes* [65]. White-rot fungi are the most studied group in relation to their ability to produce laccases to degrade lignin [63]. Indeed, in nature, fungal laccases are involved in lignin degradation and in the removal of toxic phenols produced during this process [66]. Moreover, they play a role in the synthesis of dihydroxynaphthalene melanins, compounds that are useful for protection against environmental stress [67]. Thanks to their characteristics, laccases are used in a number of industrial applications such as delignification, pulp bleaching and bioremediation processes removing toxic compounds through oxidative enzymatic coupling [64].

### 2.2. Peroxidases (EC 1.11.1)

Peroxidases (EC 1.11.1) are a group of haem containing oxidoreductases, which catalyse the oxidation of organic and inorganic compounds and the reduction of hydrogen peroxide [68]. Peroxidases are divided into three classes: class-I are intracellular peroxidases that are found in most living organisms, except animals; class-II are extracellular fungal peroxidases; class-III are extracellular plant peroxidases [69]. The main fungal peroxidases are manganese peroxidases (MnP; EC 1.11.1.13), lignin peroxidases (LiP; EC 1.11.1.14), versatile peroxidases (EC 1.11.1.16) and dye decolorizing peroxidases (EC 1.11.1.19), depending on what they use for the reducing substrate [70].

The most well-known group of peroxidase-producing fungi are the ligninolytic fungi such as white-rot fungi *Phanerochaete chrysosporium*, *Trametes versicolor*, *Pleurotus* spp., *Phlebia radiata*, *Bjerkandera adusta*, *Ceriporiopsis subvermispora* and *Dichomitus squalens* [68,70,71]. LiP and MnP were first discovered and purified in the extracellular medium of a *Phanerochaete chrysosporium* culture in the 1980s [72,73,74,75]. Since then, studies have continued to better understand the functioning of these enzymes, and in 2010 the structure of the MnP of *P. chrysosporium* was refined at 0.93 Å resolution [76].

The main characteristics of peroxidases are their non-specificity and their ability to oxidise substrates with high redox potential [70,77]. These properties have led to these enzymes being used in a large number of applications. Peroxidases are involved in the production of biofuels and paper [78], in waste treatment [68] and in the bioremediation of industrial pollutants such as synthetic dyes and polycyclic aromatic hydrocarbons (PAHs) [40,79,80,81,82,83,84,85].

### 2.3. Cutinases (E.C. 3.1.1.74)

Cutinases (E.C. 3.1.1.74) are extracellular serine esterases and are divided into two fungal subfamilies and one bacterial subfamily [86]. These subfamilies have a different primary structure according to whether they are eukaryotic or prokaryotic [87]. Cutinases have an α/β fold, and a central β-sheet composed of five parallel strands covered by two or three helices on either side of the sheet [88]. Their active site is uncovered and consists of a catalytic triad of Ser-His-Asp/Glu [89]. One exception is the active site of *Trichoderma reesei* cutinase, which has a covered active site, similar to a lipase [90]. In this case the imidazole of the histidine removes a proton from the serine hydroxyl group and the serine oxygen makes a nucleophilic attack on the substrate acyl carbonyl carbon [91]. Then, by a transacylation reaction with the serine, the substrate becomes an acyl-enzyme intermediate, which is then hydrolysed to release the product. The development of a negative charge during the formation of the acyl-enzyme intermediate is stabilized by an oxyanion hole of cutinase [88,92].

In nature, because they can degrade ester bonds, cutinases are involved in fungal pathogenesis [88]. Cutinases are also multifunctional enzymes with many industrial applications due to their ability to catalyse hydrolysis reactions, esterifications and transesterifications [93]. These properties make them suitable for the degradation of high molecular weight polyesters [94] and useful in synthetic fibre modification [95].

### 2.4. Lipases (EC 3.1.1.3)

Fungal lipases (EC 3.1.1.3) are extracellular triacylglycerol acyl hydrolases that can hydrolyse ester bonds from insoluble substrates of tri-, di- and mono-glycerides into free fatty acids and glycerol [96,97]. Therefore, lipases are involved in lipid metabolism in processes such as digestion, absorption and reconstitution [98].

Lipase-producing fungi have been isolated from different habitats such as contaminated soils, wastes and deteriorated food [98,99,100]. The main genera of lipase-producing fungi are *Aspergillus*, *Acremonium*, *Alternaria*, *Beauveria*, *Candida*, *Eremothecium*, *Fusarium*, *Geotrichum*, *Humicola*, *Mucor*, *Ophiostoma*, *Penicillium*, *Rhizomucor*, *Rhizopus* and *Trichoderma* [96,101].

Fungal lipases are used in a number of industrial applications such as in the food, textile and manufacturing industries and in the production of detergents, cosmetics and pharmaceuticals [102]. Moreover, lipases can degrade fatty wastes [103] and polyurethane (PUR) [104].

## 3. Types of Plastics and Their Biodegradation by Fungi

### 3.1. Polyethylene (PE)

Polyethylene (PE) is one of the most abundant and most widely commercialized synthetic, petroleum-based, thermoplastic materials [105,106]. PE consists of long linear chains of ethylene monomers (C_2_H_4_)_n_ [107] and can be divided into high-density polyethylene (HDPE) and low-density polyethylene (LDPE), according to the number of carbon atoms and the molecular weight [108] (Table 2). HDPE, so called because it has a molecular weight between 100,000–250,000 Daltons [109], is used in bottles for cleaning products, houseware items and for toy production, making up 12.9% of the total plastic demand in Europe [4]. On the other hand, LDPE has a molecular weight of 40,000 Daltons [109] and is mainly used in single-use plastics such as bags and food packaging films, composing 17.4% of the total plastic demand in Europe [4]. Due to the short time period for the use of PE products, they are rapidly accumulated in the environment in large numbers [110,111]. Over the years, this large abundance of PE in the environment has led to the investigation of alternative disposal methods, including microbial biodegradation.

One of the main problems of using microbes to degrade PE is its high molecular weight, which limits the number of possible enzymatic reactions. Usually, a microorganism’s enzymatic systems use substrates with 10–50 carbons. Therefore, in order to degrade PE using microbes, a reduction in its molecular weight is necessary. Moreover, the transport of molecules through the cell membrane also requires a lower molecular weight [49]. After the decrease in molecular weight, enzymes must oxidise the polymer to transform it into a carboxylic acid that can be metabolized by β-oxidation and the Krebs cycle [112]. The reduction in molecular weight can be performed using abiotic factors such as UV light and heat, or by microbial enzymes [49]. Most studies investigating this enzymatic reduction have focused on bacteria that use laccase and alkane hydroxylase from the AlkB family [113,114]. However, the advantage of fungal biodegradation is in the production of exoenzymes. Hydrophobins, fungal surface hydrophobic proteins that facilitate hyphae adhesion to a plastic surface, and exoenzymes, such as peroxidases and oxidases, promote oxidation or hydrolysis [38,39,41,115,116]. The most studied fungal genera belong to Ascomycota and are *Aspergillus*, *Penicillium, Trichoderma* and *Fusarium* [117,118,119,120,121,122,123], whereas Basidiomycota or Mucoromycota are less investigated. In addition, the involvement of fungi such as *Phanerochaete chrysosporium* [124], *Bjerkandera adusta* [125], *Trametes versicolor* [115] and *Rhizopus oryzae* [126] in PE biodegradation has also been reported.

### 3.2. Fungal Enzymes Involved in PE Biodegradation

The main fungal enzymes involved in polyethylene biodegradation are the lignolitic enzymes laccases (Lac, EC 1.10.3.2.) and peroxidases (EC 1.11.1.7) [116,117,120,127].

The effect of these enzymes on PE has been studied extensively in Basidiomycota, but they are also present in Ascomycota. For example, the ascomycete *Trichoderma harzianum* is able to produce both laccase (Mw 88 kDa) and peroxidase (Mw 55 kDa) when involved in PE biodegradation [120]. Indeed, the treatment of PE with 0.01071 IU/mL of its laccase caused a reduction in mass of 0.5% after 10 days incubation, while the treatment of PE with 0.01080 IU/mL of its peroxidase caused a 0.6% loss of mass. As a result of the enzymatic treatment, carboxylic acids, aldehydes, aromatics, alcohols, esters, ethers and alkyl halides groups were formed and detected by Fourier-transform infrared spectroscopy (FTIR) analysis.

A particularly interesting ascomycete involved in HDPE microplastic biodegradation (density 0.955 g/cm^3^, size below 200 μm) is *Aspergillus flavus* PEDX3, which was isolated from the gut of the wax moth *Galleria mellonella* [38]. This strain was able to depolymerize HDPE long chains and produce lower molecular weight fragments after 28 days of incubation. *A. flavus* PEDX3 action could be attributed to its capacity to produce laccases and laccase-like multicopper oxidases (LMCOs). Gene sequencing analysis using RT-PCR led to the identification of two genes (AFLA_006190 and AFLA_053930) that may encode potential degrading LMCOs [38].

In the Basidiomycota, partially purified manganese peroxidase (MnP, EC 1.11.1.13) from *Phanerochaete chrysosporium* ME-446 caused significant PE degradation when 0.1% Tween 80 was present in the growing medium, reducing tensile strength and elongation [115]. Moreover, after the addition of 0.1 mM manganese sulfate (MnSO_4_), PE molecular weight (Mw) decreased from 716,000 to 89,500 Daltons and the relative elongation changed from 100% to 0%. Although exogenous H_2_O_2_ supply is not necessary for polyethylene degradation, it is essential for the MnP reaction system [115].

Fujisawa et al. [128] investigated the effects of a laccase-mediator system (LMS) from *Trametes versicolor* IFO 6482 in PE biodegradation. LMS (500 nkat) was able to reduce the PE elongation by 20% in 3 days, while the addition of 0.2 mM 1-hydroxybenzotriazole (HBT) to the medium caused no elongation and relative tensile strength decreased by 60%. Moreover, Mw changed from 242,000 to 28,300 Daltons after 3 days of LMS with HBT mediator treatment at 30 °C.

Another Basidiomycete involved in PE biodegradation is *Pleurotus ostreatus,* which can hydrolyse C–C bonds by producing extracellular ligninolytic enzymes including lignin peroxidase (LiP), manganese peroxidase (MnP) and laccases (Lac). During growth on semisolid Radha modified medium in the presence of LDPE sheets, high enzyme production was detected. Specifically, the highest Lac and LiP activities were 2.817 U/g and 70.755 U/g after 30 days and 90 days, respectively, while the highest MnP production was observed at day 120 (1.097 U/g) [48].

A recent study reports computational molecular simulations between PE (dodecane, 170.3 Daltons) and different enzymes known to degrade it. Santacruz-Juárez et al. [116] studied the interactions between MnP (manganese peroxidase from *Phanerochaete chrysosprium*), LiP (lignin peroxidase from *Trametes cervine*), Lac (laccase from *Trametes versicolor*), UnP (unspecific peroxygenase from *Agrocybe aegerita*) or Cut (cutinase from *Fusarium solani*, as a negative control) and PE. They measured the binding affinity, i.e., the strength of the binding interaction between the enzyme and its ligand (PE), and found affinities were UnP (34.34 μM) > Lac (40.11 μM) > LiP (66.93 μM) > MnP (82.16 μM) > Cut (5590 μM). The high area (659.920 Å^2^), volume (367.243 Å^3^) and hydrophobicity of the UnP catalytic cavity was suggested to be the reason for the high interaction with PE [129,130,131]. The hydrophobicity is caused by the presence of phenylalanine residues in the UnP active site [116]. The binding affinity mirrored the binding energy scores, which resulted in −6.09, −6.00, −5.69, −5.57 and −3.07 Kcal/mol for UnP–PE, Lac–PE, LiP–PE, MnP–PE and Cut–PE complexes, respectively. Indeed, the lower the required binding energy, the easier the bonds are created. These computational observations showed that peroxidases can play an important role in PE biodegradation, and that the non-specific UnP enzymes can be used in practical applications due to their distinctive cavity composed of Val244, Phe121, Phe191, Phe199, Phe274, Ala77, Thr192, Gly195, Glu196, Ser123, Cys33, haem propionate, 1H-imidazol-5-yl methanol (Mzo354) and two water molecules [116].

Hypothetical biodegradation pathways involving the ligninolytic enzymes (Lac, LiP and MnP) and PE have been proposed, using as the biosurfactant a fungal hydrophobin from class-II [41,116]. Specifically, Bertrand et al. [132] hypothesized a PE degradation pathway that uses Lac from *Trametes versicolor*; Miki et al. [133] suggested using LiP from *Trametes cervine*; Sánchez [41], in a computational study, proposed using MnP from *Phanerochaete chrysosporium*. In order to perform their degradative activity, both MnP and LiP require the addition of H_2_O_2_ to the culture medium [116] and acidic conditions [41]. The involvement of H_2_O_2_ is to act as an electron accepting co-substrate in the oxidation-reduction reactions promoted by MnP and LiP [41]. Alternatively, Lac causes the transfer of electrons from organic substrates to molecular oxygen. Therefore, the main difference in PE biodegradation pathways between laccases and haem peroxidases (LiP and MnP) is based on the different methods of electron transfer [41].

### 3.3. Polyethylene Terephthalate (PET)

Polyethylene terephthalate (PET) is another important synthetic, petroleum-based, thermoplastic polymer commonly used in everyday life [134,135]. PET is a saturated polyester consisting of terephthalic acid (TPA) and ethylene glycol (C_10_H_8_O_4_)_n_ with an average molecular weight ranging from 20,000 to 50,000 Daltons depending on the field of application [136] (Table 3). PET is widely used in beverage bottles, synthetic textile fibres, films and resins, and it has a significant market share of plastic fibres thanks to its versatile performance characteristics [137,138]. PET fibres are employed in cloth, technical and medical textiles, and in furnishings [137]. Nowadays, PET represents 8.4% of the plastics demand in Europe [4].

PET depolymerisation takes place when it is converted to terephthalic or isophthalic acid, ethylene glycol and small oligomers such as bis(2-hydroxyethyl) terephthalate (BHET) and mono(2-hydroxyethyl) terephthalate (MHET) [139]. These polymers are less harmful to the environment than PET [140]. Although PET is a polyester (compounds that are considered to degrade more easily), it is recalcitrant to biodegradation [141]. However, fungal genera such as *Aspergillus* [142,143], *Fusarium* [144,145] and *Penicillium* [146,147] have been reported to be involved in PET biodegradation.

### 3.4. Fungal Enzymes Involved in PET Biodegradation

In recent times, several studies have looked for enzymes involved in PET biodegradation. However, most studies have focused on bacterial enzymes [148,149,150] with fungal enzymes being less investigated.

The main fungal enzymes involved in PET biodegradation are hydrolytic enzymes acting on ester bonds (esterases; EC 3.1.1), such as cutinases (EC 3.1.1.74), lipases (EC 3.1.1.3) and carboxylesterases (EC 3.1.1.1) [138,149,151].

#### 3.4.1. Cutinases Involved in PET Biodegradation

Specific cutinases able to degrade PET were identified from *Humicola insolens* (HiC) [148,152], *Fusarium solani pisi* (FsC) [148,152,153,154,155] and *Fusarium oxysporum* (*Fo*Cut5a) [94].

The most studied enzymes are the cutinases HiC and FsC. HiC has good thermostability with a temperature range from 30 to 85 °C, an optimum at 80 °C, and maximum initial activity from 70 to 80 °C. On the other hand, FsC has a lower temperature range of 30–60 °C with the best performance at 50 °C. Ronkvist et al. [152] tested the biodegradation capacity of HiC and FsC. They found that the hydrolysis rate constant *k*_2_ was 7-fold higher for HiC at 70 °C than FsC at 40 °C (0.62 μmol/cm^2^/h compared to 0.09 μmol/cm^2^/h) [152]. Moreover, the results showed a 97 ± 3% weight loss when low-crystallinity PET was incubated with HiC for 96 h at 70 °C, while there was only a 5% decrease after 96 h of incubation with FsC at 40 °C.

A few studies noted that the activity of cutinases was higher for an amorphous PET polymer compared to that of a highly crystalline substrate [43,154]. Indeed, these enzymes are sensitive to chain distribution and length [156]. An increase in the PET crystallinity rate from 7% to 35% caused a decrease in the initial enzymatic activities up to 25-fold for HiC and 6-fold for FsC [152]. The enzymes’ preference for amorphous regions of PET led to an increase in the biodegradation of these regions and an increase in the crystallinity rate of the biodegraded polymer [152,153]. Moreover, a high presence of aromatic rings lowers the rate of hydrolysis. On the other hand, HiC preferably hydrolysed both internal (terephthalic acid-1,4-butanediol) and external (benzoic acid-1,4 butanediol) ester bonds, and more rapidly hydrolysed substrates with longer terminal alcohols but shorter chain length acids [157].

Several studies have focused on trying to explain the biodegradation pathways and functioning of HiC and FsC. For example, the ability of cutinases to cleave ester bonds of dissolved materials [158] was confirmed by Eberl et al. [153], who studied the PET monomer bis(2-hydroxyethyl) terephthalate (BHET). They observed that a cutinase from *F. solani* was able to completely hydrolyse BHET in only 30 min. Terephthalic acid (TPA) formation started after the complete hydrolysis of BHET to the monoester mono (2-hydroxyl ethyl) terephthalate (MHET). After the 96 h incubation of low-crystallinity PET with FsC in 1 M Tris-HCl (pH 8) at 40 °C, or with HiC at 70 °C, MHET was reduced to terephthalic acid (TPA) and ethylene glycol [152,159]. Nevertheless, after FsC-catalysed PET hydrolysis, Vertommen et al. [43] observed a predominant production of MHET, some TPA and small traces of BHET. These differences could be ascribed to changes in the ratio of substrate—enzyme, or changes in the incubation conditions, which could lead to the incomplete conversion of water-soluble PET degradation products into TPA.

The active site of a cutinase from *Fusarium solani pisi* (PDB code1CEX) was genetically modified to improve its activity towards PET fibres [160]. Previous studies proposed a 3D structure of this enzyme [88] and the native cutinase gene sequence that can be PCR-amplified with the primers CutFor (5′-CGGGATCCCATGAAACAAAGCACTATTGCACTG-3′) and CutRev (5′-CGAGCTCGCAGCAGAACCACGGACAGCC-3′) from the vector pDrFST [161]. This information allowed Araújo et al. [157] to carry out computational studies showing that the increase in the size of the active site corresponds with a 4–5-fold enhancement in activity. Indeed, the large PET polymer can fit better in the active site after the amplification. Furthermore, it was shown that the mutations L81A, N84A, L182A, V184A and L189A result in a better stabilization of the tetrahedral intermediate of the model substrates [160] due to the enlargement of the active site.

Another cutinase with potential in PET bioremediation is produced by *Fusarium oxysporum*, and is called *Fo*Cut5a [94,144]. It is highly homologous to *F. solani pisi* cutinase (FsC), but the hydrophobic residues Ala62 and Phe63 present in *F. solani* are replaced by Lys63 and Tyr64 polar amino acids in *Fo*Cut5a at the end of helix a2. Due to these and other small but significant differences, *Fo*Cut5a seems slightly more thermostable than FsC, underlining a possible important role in industrial applications [94]. The optimized parameters for PET hydrolysis are 40 °C, pH 8 and 1.92 mg *Fo*Cut5a per gram of fabric [144]. *Fo*Cut5a efficacy was confirmed by superficial changes observable by Fourier-transform infrared spectroscopy (FTIR) ATR analysis, X-ray photoelectron spectroscopy (XPS), Scanning Electron Microscope (SEM), as well as through dyeability tests using reactive dyes [144].

#### 3.4.2. Lipases Involved in PET Biodegradation

Lipases are another class of enzymes involved in PET biodegradation [149]. The most studied that are involved in PET biodegradation are produced by *Aspergillus oryzae* CCUG 33812 [143] and by the yeasts *Candida antarctica* (CALB) [140,162] and *Pichia pastoris* [163].

*Aspergillus oryzae* CCUG 33812 can produce a lipase able to catalyse PET hydrolysis using 0.1 g/L bis(2-hydroxyethyl) terephthalate (BHT) as an inducer [143]. An increase in hydrophilicity and antistatic ability, as well as a 0.74% weight loss and a decrease in both the water contact angle and static half decay time, were observed after 24 h at 55 °C [143].

The lipase triacylglycerol hydrolase produced by the yeast *Pichia pastoris* was able to modify the surface morphology of polyester fibres at 60 °C and at pH 7.5–8. Moreover, 7 h treatment with the combination of 10 g/L *P. pastoris* lipase and 0.5 g/L non-ionic surfactant JFC (a fatty alcohol polyoxyethylene ether) at 60 °C and pH 7.5 changed the surface morphology of the PET fibres and increased the number of hydrophilic groups [163].

#### 3.4.3. Polyesterases Involved in PET Biodegradation

Extracellular polyesterases involved in PET hydrolyzation are secreted by *Beauveria brongniartii* [164] and *Penicillium citrinum* grown on a medium containing cutin with molecular weight 14.1 kDa, temperature optimum 36 °C and pH 8.2 [165].

Polyesterase from *B. brongniartii* released TPA during treatment of PET [164], while *P. citrinum* enzymatic activity liberates only low amounts of TPA in favour of BHET and MHET [165].

#### 3.4.4. Synergic Action of Cutinase HiC and Lipase CALB

An important PET depolymerization enzyme that can act synergically with HiC is lipase B from *Candida antarctica* (CALB) [140,162].

HiC and CALB present two different activity profiles at the final stage of PET depolymerization. Indeed, TPA was the predominant molecule after 24 h of CALB action, while HiC very quickly converted BHET into MHET, but then TPA formation was slow [140]. Despite this, when the two enzymes are used alone, HiC is more efficient than CALB in degrading PET [149], while a synergic action between HiC and CALB led to a more intense MHET consumption and TPA formation [159]. Better results were obtained by de Castro et al. [140] using HiC and CALB sequentially. This method led to an initial release of MHET (HiC action at 60 °C), which was rapidly converted to TPA after CALB addition (37 °C), resulting in a degradation 141-times higher than when using the two enzymes at the same temperature [140].

#### 3.4.5. PET Hydrophobicity Modification after Fungal Enzymes Action

An interesting reaction observable with most enzymes involved in PET hydrolysation is the increase in hydrophilicity. The action of hydrolases from *Fusarium oxysporum* LCH1 [145] leads to a higher PET hydrophilicity than cutinase from *F. solani pisi* [137,166,167,168]. Indeed, the water adsorption of PET fabrics treated with 80 U of cutinase and hydrolase was 36 mm/10 min and 57 mm/10 min, respectively [145]. An increase in hydrophilicity was also observed after the action of polyesterase obtained from *Penicillium citrinum* [165]. Comparing the measure of rising height, the enzyme preparation of *P. citrinum* caused the largest increase in hydrophilicity on PET (5.1 cm) [165], followed by *Aspergillus* sp. (4.3 cm), *F. solani* (2.3 cm) and *Beauveria* sp. (1.2 cm) [155]. The increase in hydrophilicity could be caused by the introduction of polar groups onto the polymer surface [167,168]. This was confirmed by the increase in hydrophilicity associated with the gain of hydroxylic and carboxylic acid groups on the PET surface after hydrolysis by an enzyme preparation from *Beauveria brongniartii* (0.5 nKat/mL) [164].

### 3.5. Polyurethane (PUR)

Polyurethane (PUR) is a polymeric, synthetic, petroleum-based, thermosets polymer composed of repeating units containing ethyl carbamate (H_2_NCO_2_C_2_H_5_) and urethane bonds with urea, ether, ester and aromatic groups [169]. Depending on the polyols used in the condensation reaction to produce PUR, polyurethane can be classified as a polyester (esters were used) or a polyether (utilisation of ethers) [47] (Table 4). PUR is a very versatile material, which can be used as a foam in building insulation, in pillows and mattresses, in semirigid plastics, as an elastomer, and in paint, adhesive and textile fibres [170,171]. As with PET, polyurethane made up 7.8% of European plastic demand in 2020 [4].

Today the disposal of polyurethane is a major problem as its degradation takes hundreds of years [172,173]. Moreover, incineration generates toxic emissions, and the material becomes useless after a few rounds of recycling [174].

The effectiveness of fungal biodegradation of PUR depends on the polymer molecular orientation, cross-linking and crystallinity [175]. It is easier to attack polyurethane in the amorphous regions rather than in the crystalline regions [169]. Moreover, the availability depends on the absence of ramifications or on the presence of long repetitive units that decrease the formation of highly crystalline areas [176,177]. In addition, the presence of methylene groups encourages fungal degradation [177].

Depending on its composition, polyurethane can be classified as a polyester or a polyether polyurethane, which can be more or less targeted by fungi [177]. The ester types are more susceptible to microbial degradation than ether types [178]. Indeed, during biodegradation, polyesters are subjected to hydrolysis, a reaction catalysed by both acids and bases and which follows the three-centre mechanism. The result of the hydrolysis of ester bonds is the liberation of a free acid, making this reaction autocatalytic [173]. On the other hand, ether bonds are very stable and as a result ether-PU products are a serious problem in environmental pollution [179]. Biodeterioration of polyether polyurethane can be carried out by oxidation rather than hydrolysis, and it involves the abstraction of α-hydrogen adjacent to oxygen [180,181]. Adding metal ions such as cobalt can accelerate this type of degradation [173].

Fungal biodegradation of ether-PU appears to be more promising than bacterial, especially in foams. This could be due to the ability of fungal hyphae to penetrate into foam pores, cracking the material mechanically [182] and thereby increasing the accessibility of chemical bonds for fungal exoenzymes. On the other hand, ester-PU is a substrate that is more susceptible to fungal enzymes than ether-PU, and several ester-PU degrading fungi are reported in the literature [183].

### 3.6. Fungal Enzymes Involved in PUR Biodegradation

Fungal hydrolases such as proteases, esterases, ureases and lipase are involved in the polyurethane biodegradation process. Ester bonds are cleaved by all these enzymes, while amide and urethane bonds are only hydrolysed by proteases, and urea is only targeted by ureases. Urease activity can be detected by phenolic compounds released into culture media, and this activity is exploited to estimate the amount of biodegradation [177]. However, the action of degradative enzymes is susceptible to the distance between urethane bonds, which can interfere negatively in the biodegradation [183]. There are also specific enzymes able to degrade polyurethane, and they are known as polyurethanases (PUase) [39,184].

#### 3.6.1. Esterases Involved in Polyester-PUR Biodegradation

The mechanism of polyester-PUR degradation by esterases was proposed for the first time by Wales and Sagar in 1988 [185]. They suggested that extracellular esterases act by hydrolysis on polymers containing ester and urethane links. Boubendir [186] reported the correlation between the increase in esterase activity and the addition of liquid polyester PUR in cultural medium, suggesting the induction of enzymes with esterase and urethane hydrolase activities in *Chaetomium globosum* and *Aspergillus terreus*. Another extracellular enzyme-like factor with esterase properties was secreted by *Curvularia senegalensis* [185]. This enzyme was found to be stable at 100 °C for 10 min, displayed a band of 28 kDa in SDS PAGE analysis and was inhibited by phenylmethylsulphonyl fluoride (PMSF). Its involvement in polyurethane degradation was observed through the formation of clear zones in Impranil agar plates [185]. Impranil^®^ DLN (Impranil, Leverkusen, Germany) is a class of plastics belonging to the polyurethane family [187], and it is commonly used to determine the ability of a microorganism or a protein to degrade PUR [188].

The esterase ability of PU biodegradation was confirmed by the 1500-fold increase in esterase production by *Cladosporium pseudocladosporioides* T1.PL.1 using Impranil as the sole carbon source [47], and by the increase in *Aspergillus fumigatus* strain S45’s esterase activity from 0.1440 µM/Min/mg to 2.687 µM/Min/mg after 15 days of incubation with polyester PUR as the sole carbon source [189].

Another fungal enzyme able to biodegrade Impranil DLN is produced by *Pestalotiopsis microspore* E2712A. This observation suggests the involvement of the 21 kDa serine hydrolase-like enzyme in PU biodegradation [190].

#### 3.6.2. Lipases Involved in Polyester-PUR Biodegradation

Lipases involved in the hydrolytic degradation of polyester polyurethane are secreted mainly by yeasts belonging to the *Candida* genus. Kinetics and mathematical models of PUR degradation by *Candida rugosa* were proposed by Gautam et al. [191]. In this study, the maximum degradation rate of 2.5 g PUR/L was reached using 70 μg lipase/mL at pH 7 and 35 °C. Moreover, a linear increase in the PUR degradation product, diethylene glycol, was observed during PUR biodegradation [191].

*Candida antarctica* lipase represents another potential candidate for PUR biodegradation [104]. Indeed, oligomers with a molecular weight lower than 500 Daltons were produced after the activity of 20 mg of lipase in toluene (10 mL) at 60 °C for 24 h [104].

The Ascomycete *Aspergillus tubingenesis* is able to degrade polyester PU, producing both esterases and lipases [192]. The optimum pH required for *A. tubingenesis* esterase maximum activity was pH 7, while lipase has an optimum at pH 5. For both enzymes, the optimum temperature was 37 °C. An interesting observation made by the authors of this study is that lipase production augments in the first two weeks of culture when the surfactants Tween 20 or Tween 80 were added to the culture medium, then the lipase concentration decreased by the end of the month. On the other hand, esterase activities reached the maximum activity after two weeks in the presence of Tween 80 and continued to increase slowly until the fourth week in the presence of Tween 20. The gradual decrease in enzyme production was hypothesized to be due to the surfactants’ repression effect on the gene responsible for enzyme production [192].

#### 3.6.3. Cutinase Involved in Polyester-PUR Biodegradation

A cutinase demonstrating the ability to degrade PUR was secreted by the thermophilic fungus *Thielavia terrestris* CAU709 [193]. The highest production of this enzyme (TtcutA) was observed when *T. terrestris* CAU709 grew using 2% (*w*/*v*) wheat bran as the sole carbon source amended with 0.1% (*w*/*v*) Tween 80. The highest enzyme activity (90.4 U/mL) was observed after 96 h of cultivation. The molecular mass of the purified TtcutA was between 25.3 and 22.8 kDa with optimum conditions at pH 4.0 and a temperature of 50 °C [193]. PU biodegradation was estimated using 0.6% emulsified PU [193].

#### 3.6.4. Fungal Hydrolyses Involved in Polyether-PUR Biodegradation

As previously reported, enzymatic biodegradation of polyether PUR is more complex than polyester PUR and only a few studies have reported its degradation by fungal hydrolyses [151,169,194]. Urethane-bond-degrading enzymes and urea-bond-degrading enzymes were found in a culture supernatant of *Alternaria* sp. PURDK2. Polyamines (4,4′-methylenedianiline) and polyols were released after the hydrolysation of urethane bonds, while 4,4′-methylenedianiline and *n*-polyamines were generated after the degradation of urea bonds by enzymes of *Alternaria* sp. PURDK2 [179]. However, an initial mechanical disruption by fungal hyphae was required for polyether-PUR biodegradation [179].

### 3.7. Polyvinyl Chloride (PVC)

Polyvinyl chloride (PVC) is a commonly used synthetic, thermoplastic, petroleum-based material [195]. Pure PVC consists of long chains of ethylene monomers containing 56.77% (*w*/*w*) chlorine element (C_2_H_3_Cl)_n_ [193] (Table 5). It is a white, brittle, solid polymer, is highly hydrophobic, soluble in tetrahydrofuran (THF) and resilient to chemical abrasion [24,196].

PVC represents 9.6% of the total European plastic use [4]. It is used in its rigid or flexible form in pipes and electrical wire insulation, profiles of windows or doors, coverings for floors and walls, and in textiles such as synthetic leather products, shoe soles, packaging and credit cards [24,170,196].

Some PVC products have a long lifespan so there is a long time between PVC production and waste, but in some products, such as packaging, PVC has a short life. In any case, long- and short-term PVC products are disposed of at the end of their use [195]. Depositing in landfill and incineration are the most widely used disposal methods for PVC [196]. However, during incineration a large amount of hydrogen chloride and tetrachlorodibenzo-p-dioxin are produced, causing secondary pollution [195,196,197,198]. Therefore, PVC biodegradation has become a topic of particular interest.

The biodegradation of PVC involves three main reactions: chain depolymerization; oxidation; mineralisation of formed intermediates [196,197]. Fungal PVC biodegradation is a very difficult process due to the hydrophobicity of the material, its resistance to abrasion and the persistence of its structure [199]. Indeed, modification of the molecular weight of PVC by microorganisms has been reported only in a small number of studies [196]. For example, fungal involvement in PVC biodegradation has been found in white-rot fungi [200] such as *Pleurotus* sp., *Polyporus versicolor*, *Phanerochaete chrysosporium* and *Lentinus tigrinus* [201,202]; or by Ascomycetes belonging to the genera *Aspergillus* [202,203,204,205,206,207,208], *Cochliobolus* [209], *Chaetomium* [203,210,211], *Fusarium* [203,205], *Mucor* [203,212] and *Penicillium* [203,205,206,212,213].

### 3.8. Fungal Enzymes Involved in PVC Biodegradation

The fungal enzymes involved in PVC biodegradation are currently not well known [178,214,215,216].

It has been reported that PVC biodegradation is related to the ability to degrade lignin [196,217], as demonstrated by the modification of the PVC structure by fungal lignin peroxidase (EC 1.11.1.14) from *Phanerocheate chrysosporium* [218]. *P. chrysosporium* lignin peroxidase had a molecular weight of 46 kDa and reached its maximum production after 4 weeks at 25 °C and pH 5. The weight of PVC films decreased by 31% when this partially purified enzyme was used and a stretch of alkenyl C–H in the PVC structure was observed by FTIR analysis (peak at 2943 cm^−1^) [218].

Another ligninolytic enzyme involved in PVC biodegradation is laccase from *Cochliobolus* sp. [219]. The higher laccase production (1.793 nKat/mL) was observed after 6 days of incubation at 30 °C and at pH 6.5. Modifications in FTIR spectra, such as the shifting of the CH-stretching mode (from 2912 cm^−1^ to 2915 cm^−1^) and the appearance of new peaks corresponding to a carbonyl group, suggested there was activity of *Cochliobolus* sp. laccase on the PVC structure. Changes in the PVC structure and surface were confirmed by GC-MS analysis and SEM photography [219].

Other studies have reported the involvement of fungal ligninolytic enzymes in PVC biodegradation [201]; however, no attempt was made to quantify the enzyme production or to demonstrate substrate degradation by purified enzymes [220].

### 3.9. Polypropylene (PP)

Polypropylene (PP) is a linear, thermoplastic hydrocarbon synthetic polymer, where a methyl-group replaces one hydrogen for every carbon of PE structure (C_3_H_6_)_n_ [35,221]. PP is an inert, semicrystalline material, slightly harder and more resistant to heat and chemical reactions than PE [222]. It has a high hydrophobicity [223] and a molecular weight varying from 10,000 to 40,000 Daltons [214].

Polypropylene is the most used plastic polymer in Europe, accounting for 19.7% of the plastic demand [4]. It is used in many industries such as packaging for food and materials, pipes, automotive parts and diapers [32] (Table 6).

Due to the short-term use of these packaging products and the massive and rapid accumulation of PP in the environment, the study of alternative disposal methods, such as fungal biodegradation, is an important field of research.

Polypropylene proprieties such as hydrophobicity and high molecular weight make it particularly resistant to microbial attack and biodegradation [214]. Moreover, the addition of stabilizers and antioxidants to preserve PP from oxidation by atmospheric agents increase its biodegradation resistance even more [224]. Despite these problems, fungi such as *Aspergillus niger* were found to be involved in PP biodegradation [225,226,227]. Only a few other fungal species such as *Lasiodiplodia theobromae* [228], *Bjerkandera adusta* [229] and *Engyodontium album* [228] were reported as being able to attack PP. Due to its high resistance and recalcitrance, most studies used pretreatments such as UV or γ-irradiation to make PP more susceptible to degradation by fungi [226,228,230].

### 3.10. Fungal Enzymes Involved in PP Biodegradation

Although some studies reported the involvement of fungi in polypropylene biodegradation, so far none have investigated the use of fungal enzymes in the biodegradation of PP [101,178,214,216,231,232]. Investigating which microorganisms are able to degrade PP and the identification of their enzymes would be an interesting and useful area of further research.

### 3.11. Polystyrene (PS)

Polystyrene (PS) is a very stable, synthetic, thermoplastic polymer formed by an aromatic styrene monomer (C_8_H_8_)_n_ [216,233]. PS has a large molecular weight and high hydrophobicity, which contribute to its rigidity and hardness [214]. The low cost and ease of production, as well as its lightweight, rigid and transparent properties make it ideal for the packaging industry [219,233]. PS is also used in electrical and electronic equipment, and in construction applications [170,234], composing 6.1% of the plastics demand in Europe [4] (Table 7).

Due to the high resistance of the C–C backbone [235], only a few studies have reported fungi involved in PS biodegradation. The main fungal genera involved in PS modification are the Ascomycetes *Rhizopus* [236], *Aspergillus* [236], *Penicillium* [237] and *Curvularia* [238]; the Basidiomycetes white-rot fungi *Phanerochaete chrysosporium* [236,239], *Trametes versicolor* [239] and *Pleurotus ostreatus* [239]; and the brown-rot fungus *Gloeophyllum trabeum* [44].

### 3.12. Fungal Enzymes Involved in PS Biodegradation

Although some studies have reported fungi involved in PS biodegradation, research describing fungal enzymes involved in this process are scarce [25,215]. Only an esterase produced by *Lentinus tigrinus* has been reported to biodegrade PS film into non-toxic smaller molecules [240]. The optimum activity of this purified hydrolytic enzyme was observed at 45 °C and pH 9. Moreover, the maximum production reached was 38.62 U/mL when *L. tigrinus* was grown in the presence of urea and yeast extract [240].

## 4. Conclusions and Perspectives for Future Research

Plastic waste is an increasingly urgent environmental problem, and a large amount of scientific research has been focused on using microbes for its biodegradation. In this context, the use of fungal enzymes as bioremediation tools is being increasingly studied. This review has reported the main fungal enzymes involved in plastic degradation, describing their characteristics and efficiency (Table 8). Fungal ligninolytic enzymes (laccases and peroxidases), already used in several biotechnological applications, were found to be successful in partially degrading PE and PVC. Moreover, fungal esterases, such as cutinase and lipase, were able to degrade PET and PUR. In addition, protease and urease can also contribute to PUR degradation. On the other hand, there is an absence of studies on fungal enzymes able to act on PP and PS, which opens the door for further research in this area.

Biochemical methods and tools can be developed for controlling and improving enzyme catalytic activities and could be applied in several sectors including green chemistry [241]. For example, enzyme immobilization is a powerful tool that allows the enzyme of interest to be confined in a matrix, usually consisting of inert polymers and inorganic materials [241]. This technique enhances catalytic activity and increases the stability, specificity and selectivity of the immobilized enzymes, thus significantly reducing production costs [242]. A number of studies have reported that the immobilisation of PET-degrading bacterial enzymes enhances their catalytic activity and thermal stability [243,244,245], but few studies have focused on fungi [246]. Su et al. [247] immobilised cutinase from *Aspergillus oryzae*, *Humicola insolens* and *Thielavia terrestris* on the macroporous support Lewatit VP OC 1600 obtaining immobilization yields higher than 98%. Enzyme immobilisation is currently used in the treatment of dye-based wastewater from industries [248], anthracene degradation [249] and decomposition of the insecticide methyl parathion [250].

A recently emerged research approach to plastic biodegradation by fungal enzymes is the genetic engineering of fungi involving recombinant DNA technology, gene cloning, manipulation and modification [251]. Such techniques can be used to increase yields and improve the kinetics of the enzymes produced [252,253]. In the case of bacterial enzymes, improvement in the PET degradation rate was accomplished by genetic engineering, which enhanced the enzyme thermostability [141,254,255]. Such results were obtained by reinforcing the binding of the substrate to the active site [256,257,258], or by improving the substrate/enzyme surface interaction [259,260]. If similar manipulations could be carried out of genes coding for key plastic-degrading enzymes, improvements in fungi biodegradation potential could be achieved. For example, the mutations L81A, N84A, L182A, V184A and L189A enlarged the active site of the cutinase from *Fusarium solani pisi* leading to an improved fitting of the large polymer chains of PET. This modification resulted in a fivefold increase in this cutinase activity [170]. Janatunaim and Fibriani [261] were able to express the genes of the bacterium *Ideonella sakaiensis* 201-F6 coding for the enzyme monohydroxyethyl terephthalate hydrolase, which is involved in PET biodegradation, in *Escherichia coli* BL21. A similar genetic engineering process was reported for the laccase gene *FoLacc5* from *Fusarium oxysporum* cDNA. After its expression in *Pichia pastoris* X33, this methylotroph yeast was able to easily degrade synthetic blue dyes [262]. Other fungal laccase genes, for example lac-En3–1 in *Ganoderma* sp. [263] or genes for Lac in *Aspergillus* sp. [264], were expressed in *Pichia pastoris* GS115 and then used in dye degradation. Although this technique for plastic degradation is still at the pioneer stage, it appears to produce promising results and has very high potential [251].

A further step towards biotechnological use of fungal enzymes for plastic degradation could be achieved using gene editing tools [251]. The manipulation of specific, desired genes by genome editing allows the loss or gain of a function of interest [265]. However, very few studies of this kind have been carried out. One example increased the PETase activity by the inclusion of two conserved residues of the gene coding cutinases from *Thermobifida fusca* at the polyesterase active site of the bacterium *Ideonella sakaiensis* [256]. Therefore, it appears there is a very high potential of this tool to open new avenues for future research in the area of the genetic editing of fungal enzymes for plastic degradation.

In conclusion, fungal enzymes show high potential in the fight against plastic waste thanks to their broad biodegradation ability. Currently, most studies are still performed in vitro under laboratory conditions. Further research is required to better understand the mechanisms of action of these enzymes and the genetics behind them. These interdisciplinary investigations could lead to optimised fungal cell factories with high degradation efficiency and vast industrial applications to help reduce plastic waste pollution and improve the environment.

## Figures and Tables

**Table 1 microorganisms-10-01180-t001:** Reaction schemes of the main fungal enzymes involved in plastic degradation.

Enzyme	Enzyme Commission (EC) Number	Activity	Reaction Scheme
Laccases	1.10.3.2	Oxidoreductases	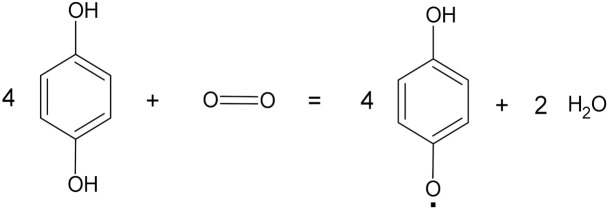 4 benzenediol + O_2_ = 4 benzosemiquinone + 2 water
Manganese peroxidases	1.11.1.13	Oxidoreductases	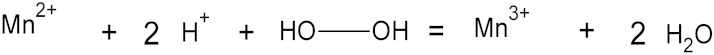
Lignin peroxidases	1.11.1.14	Oxidoreductases	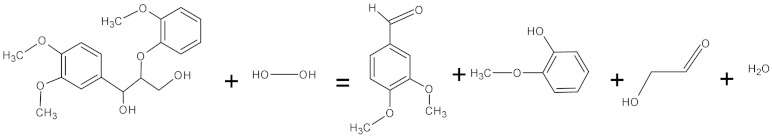 1-(3,4-dimethoxyphenyl)-2-(2-methoxyphenoxy)propane-1,3-diol + hydrogen peroxide = 3,4-dimethoxybenzaldehyde + 2-methoxyphenol + glycolaldehyde + water
			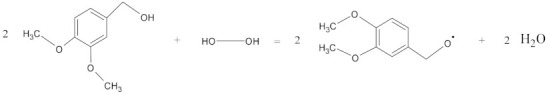 2 (3,4-dimethoxyphenyl)methanol + hydrogen peroxide = 2 (3,4-dimethoxyphenyl)methanol radical + 2 water
Versatile peroxidase	1.11.1.16	Oxidoreductases	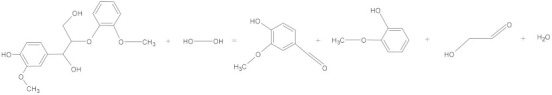 1-(4-hydroxy-3-methoxyphenyl)-2-(2-methoxyphenoxy)propane-1,3-diol + hydrogen peroxide = 4-hydroxy-3-methoxybenzaldehyde + 2-methoxyphenol + glycolaldehyde + water
Versatile peroxidase	1.11.1.16	Oxidoreductases	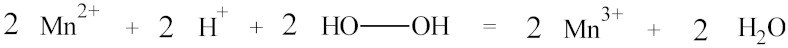
Lipases	3.1.1.3	Hydrolases	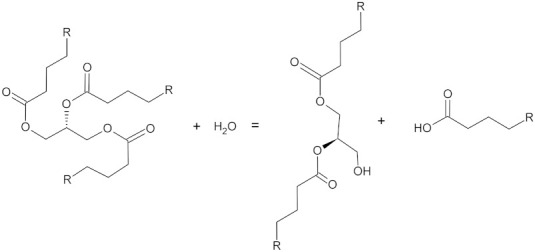 Triacylglycerol + water = diacylglycerol + a carboxylate
Cutinases	3.1.1.74	Hydrolases	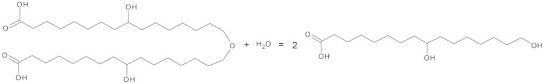 Cutin + water = 2 cutin monomers

**Table 2 microorganisms-10-01180-t002:** European plastic demand, molecular weight (Daltons) and backbone structural formula of polyethylene, high-density polyethylene and low-density polyethylene.

Name	Plastic Demand in Europe	Molecular Weight (Daltons)	Structure
High-density polyethylene (HDPE)	12.9%	100,000–250,000	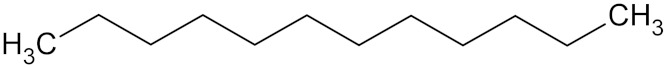
Low-density polyethylene (LDPE)	17.4%	40,000	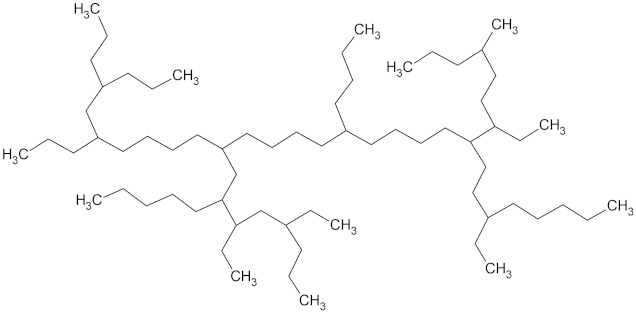

**Table 3 microorganisms-10-01180-t003:** European plastic demand, molecular weight (Daltons) and backbone structural formula of polyethylene terephthalate.

Name	Plastics Demand in Europe	Molecular Weight (Daltons)	Structure
Polyethylene terephthalate (PET)	8.4%	20,000–50,000	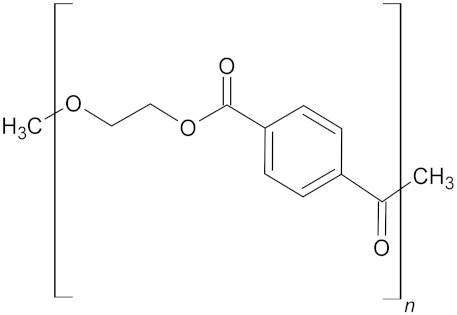

**Table 4 microorganisms-10-01180-t004:** Backbone structural formula of polyurethane (PU), ether-PU and ester-PU.

Name	Structure
Polyurethane (PU)	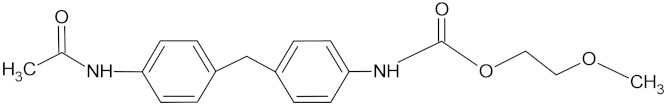
Ether-PU	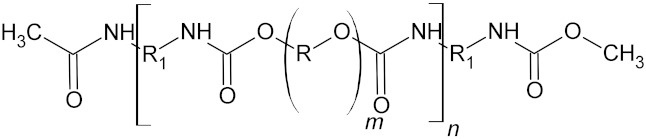
Ester-PU	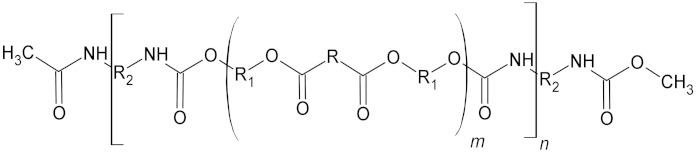

**Table 5 microorganisms-10-01180-t005:** European plastic demand and backbone structural formula of polyvinylchloride.

Name	Plastics Demand in Europe	Structure
Polyvinylchloride (PVC)	9.6%	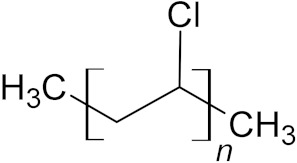

**Table 6 microorganisms-10-01180-t006:** European plastic demand, molecular weight (Daltons) and backbone structural formula of polypropylene.

Name	Plastics demand in Europe	Molecular Weight (Daltons)	Structure
Polypropylene (PP)	19.7%	10,000–40,000	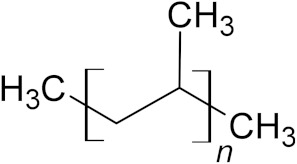

**Table 7 microorganisms-10-01180-t007:** European plastic demand, molecular weight (Daltons) and backbone structural formula of polypropylene.

Name	Plastics Demand in Europe	Molecular Weight (Daltons)	Structure
Polystyrene (PS)	6.1%	150,000–400,000	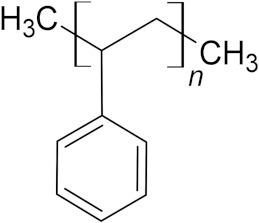

**Table 8 microorganisms-10-01180-t008:** Fungal enzymes documented as able to degrade plastics and the relative producing fungal species.

Enzymes	Fungal Species	Plastic Polymers	References
Cutinase HiC	*Humicola insolens*	PET	[148,152,157]
Cutinase *Fo*Cut5a	*Fusarium oxysporum*	PET	[94,144]
Cutinase FsC	*Fusarium solani pisi*	PET	[43,148,151,152,153,154,155,159,160]
Cutinase TtcutA	*Thielavia terrestris* CAU709	PUR	[193]
Esterases	*Aspergillus fumigatus* S45	PUR	[189]
	*Aspergillus terreus*	PUR	[186]
	*Aspergillus tubingenesis*	PUR	[192]
	*Chaetomium globosum*	PUR	[186]
	*Cladosporium pseudocladosporioides* T1.PL.1	PUR	[47]
	*Curvularia senegalensis*	PUR	[185]
Laccases	*Aspergillus flavus* PEDX3	PE	[38]
	*Pleurotus ostreatus*	PE	[48]
	*Trichoderma harzianum*	PE	[120]
Laccase-like multicopper oxidases (LMCOs)	*Aspergillus flavus* PEDX3	PE	[38]
Laccase-mediator system (LMS)	*Trametes versicolor* IFO 6482	PE	[128]
Lipases	*Aspergillus oryzae* CCUG 33812	PET	[143]
	*Aspergillus tubingenesis*	PUR	[192]
	*Candida antarctica* (CALB)	PET	[140,162]
	*Candida antarctica*	PUR	[104]
	*Candida rugosa*	PUR	[191]
	*Pichia pastoris*	PET	[163]
Peroxidases	*Trichoderma harzianum*	PE	[120]
Lignin peroxidases	*Pleurotus ostreatus*	PE	[48]
Manganese peroxidases	*Phanerochaete chrysosporium* ME-446	PE	[115]
	*Pleurotus ostreatus*	PE	[48]
Polyesterases	*Beauveria brongniartii*	PET	[164]
	*Penicillium citrinum*	PET	[165]
Serine hydrolase-like enzyme	*Pestalotiopsis microspora* E2712A	PUR	[190]
Urethane hydrolases	*Aspergillus terreus*	PUR	[186]
	*Chaetomium globosum*	PUR	[186]

## Data Availability

Not applicable.

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
