# Peer review of "Fungal Enzymes Involved in Plastics Biodegradation"

_microorganisms, 2022, doi:10.3390/microorganisms10061180_

Round 1

Reviewer 1 Report

Please see the attached comments.

Author Response

Reviewer 1

Comments and suggestions for Authors Manuscript

Title: Fungal Enzymes Involved in Plastics Biodegradation

Manuscript ID: microorganisms-1740639

The manuscript written by Marta Elisabetta Eleonora Temporiti et al. is very interesting. In this review article, fungal enzymes involved in plastics biodegradation were reviewed, future research on this area was also provided. Due to the widespread contamination of plastics and their toxic effects on human beings, there is a need to conduct the studies related to removal process of these compounds. The data presented in this manuscript will be of great interest to the readers and extend to a high degree of our knowledge about the fungal enzymes involved in plastics biodegradation.

In general, this paper is clearly laid out, very well planed and easy to read. Some specific suggestions or questions are listed below:

  1. Introduction “The introduction should briefly place the study in a broad context and highlight why it is important. It should define the purpose of the work and its significance. The current state of the research field should be carefully reviewed, and key publications cited. Please highlight controversial and diverging hypotheses when necessary. Finally, briefly mention the main aim of the work and highlight the principal conclusions. As far as possible, please keep the introduction comprehensible to scientists outside your particular field of research. References should be numbered in order of appearance and indicated by a numeral or numerals in square brackets— e.g., [1] or [2,3], or [4–6]. See the end of the document for further details on references. ” Please delete this section.

Answer: We deleted this section

  1. Introduction is easy to read but needs a little completed. Plastics have been investigated for biodegradation in many microorganisms and their enzymes. I think it is favorable to add more information into this section and cite the recent research into the field based on the literature available and described in original paper (such as doi: 10.1038/s41586-022-04599-z; doi: 10.1038/d41586-022-01075-6; doi: 10.1126/science.aba9475; doi:10.1016/j.jhazmat.2021.128033; doi:10.1016/j.jhazmat.2021.126618). This way the authors will demonstrate that they really have a good knowledge of the related literature. In addition, the novelty and significance of the manuscript were not highlighted in the Introduction section, please modify the introduction more clearly.

Answer: We added some of the suggested articles in lines 78-89. Moreover, we added a sentence to highlight the novelty and significance of our work in lines 128-130

  1. Conclusions: Authors can add and revise this section for the better understanding of the topic and its future research

Answer: We revised this section, by merging it with the section “Perspectives for future research”, creating a new section called “Conclusions and perspectives for future research”, where we tried to convey the topic better, by reorganizing and shorten it.

Reviewer 2 Report

Dear Authors

A very comprehensive review - almost no changes requested:

- The micobes/fungi and their enzymes are introduced extensively. But, how do those fungi sense for th presence of plastics to use it as a source of energy for example? and induce gene expression; or are these genes consecutively expressed? can you elaborate in few sentences on this. (some details are given for selected examples but no rule of thumb.)

- some highlights in reference list need to be finally removed.

Author Response

Dear Authors

A very comprehensive review - almost no changes requested:

- The micobes/fungi and their enzymes are introduced extensively. But, how do those fungi sense for th presence of plastics to use it as a source of energy for example? and induce gene expression; or are these genes consecutively expressed? can you elaborate in few sentences on this. (some details are given for selected examples but no rule of thumb.)

Answer: The ability of plastics to induce fungal genes involved in biodegradation is still unknown. However, it is possible to induce the production of enzymes involved in plastic degradation using substances inducers enzymes productions. For example, wood chips induce oxidoreductases production in white-rot fungi, while cutin and its monomers are inducers for cutinases in filamentous fungi. https://doi.org/10.1016/j.biotechadv.2021.107770

- some highlights in reference list need to be finally removed.

Answer: We removed every highlight

Reviewer 3 Report

1. Title.

The title is appropriate.

2. Abstract.

The abstract is appropriate.

A large group of the target audience for this article are practicing biotechnologists. Therefore, the abstract can be strengthened by indicating the groups of enzymes recommended for the degradation of specific plastics.

3. Key Words.

The key words are normal. The word “pollution” is superfluous: it does not refer to the review itself, but to the background situation.

4. Section 1. Introduction.

The introduction is normal.

There is a kind of English joke too (line 29): “able to be moulded” is not only "fit for molding, capable of being molded into various forms; pertaining to molding" but also is associated with an idea “can be under effects of moulds”.

Attention! Please, delete lines 115-123 which are some copy of the journal recommendations!

5. Section 2.

Lines 126-128. To be more punctual and in a good accordance with your references, please, replace “in nature they are required in” for: “in nature and industry they are essential in”.

6. Section 3.

This section is the most important part of the review as it not only lists biodegradable plastics but also shows the pathways of biodegradation. To my regrets, it is limited with short descriptions of the enzymatic approaches. The readers have to seek more detailed descriptions via references. It is normal for any review paper but could you kindly supplement the text with more details? For example, could you after your sentence (lines 571-572) “Biodegradation of PVC involves three main reactions: chain depolymerization; oxidation; and mineralisation of formed intermediates [193]” present several equations of chemical reactions as an illustration? Two or three such illustrative examples in the review may enhance its assistance to readers. 

7. Section 4 “Perspectives…” and Section 5 “Conclusions”.

These sections contain too many redundant common statements and hopes. For example, lines 656-663 can be removed without affecting the review. By this reason, I propose to merge these two sections into one and shorten it a bit.

8. In nutshell, the article may be accepted with the minor revision.  

Author Response

Reviewer 3

  1. Title.

The title is appropriate.

  1. Abstract.

The abstract is appropriate.

A large group of the target audience for this article are practicing biotechnologists. Therefore, the abstract can be strengthened by indicating the groups of enzymes recommended for the degradation of specific plastics.

Answer: This information is reported at line 16-17

  1. Key Words.

The key words are normal. The word “pollution” is superfluous: it does not refer to the review itself, but to the background situation.

Answer: We removed the keyword “pollution”

  1. Section 1. Introduction.

The introduction is normal.

There is a kind of English joke too (line 29): “able to be moulded” is not only "fit for molding, capable of being molded into various forms; pertaining to molding" but also is associated with an idea “can be under effects of moulds”.

Answer:  We modified the sentence as “The word plastic derives from the Greek “plastikos”, meaning “able to be modeled” (Line 29)

Attention! Please, delete lines 115-123 which are some copy of the journal recommendations!

Answer: We deleted this section

  1. Section 2.

Lines 126-128. To be more punctual and in a good accordance with your references, please, replace “in nature they are required in” for: “in nature and industry they are essential in”.

Answer: We changed the sentence as suggested

  1. Section 3.

This section is the most important part of the review as it not only lists biodegradable plastics but also shows the pathways of biodegradation. To my regrets, it is limited with short descriptions of the enzymatic approaches. The readers have to seek more detailed descriptions via references. It is normal for any review paper but could you kindly supplement the text with more details? For example, could you after your sentence (lines 571-572) “Biodegradation of PVC involves three main reactions: chain depolymerization; oxidation; and mineralisation of formed intermediates [193]” present several equations of chemical reactions as an illustration? Two or three such illustrative examples in the review may enhance its assistance to readers. 

Answer: It is difficult to find precise descriptions of reaction processes in plastic polymer biodegradation. For example, the three reactions of PVC biodegradation are not reported in detail in the cited article (196). Even after deep targeted bibliographic researches, it was not possible to find details on the reactions of depolymerization and mineralization of PVC. However, PVC oxidation process is reported in literature (197). Since it was the only step present of the whole process, we thought that showing only that reaction could give rise to doubts, so we preferred to add just the reference at line 25.

  1. Section 4 “Perspectives…” and Section 5 “Conclusions”.

These sections contain too many redundant common statements and hopes. For example, lines 656-663 can be removed without affecting the review. By this reason, I propose to merge these two sections into one and shorten it a bit.

Answer: We deleted the lines 698-705, as suggested. Moreover, we merged the two sections into the section called “Conclusions and perspectives for future research”, and we shortened it slightly.

  1. In nutshell, the article may be accepted with the minor revision.  

Round 2

Reviewer 1 Report

The authors have considered all comments raised by the reviewers and revised the manuscript accordingly based on these comments. The revision is fine and can be accepted for publication in current form.